# The TLR4/NFκB-Dependent Inflammatory Response Activated by LPS Is Inhibited in Human Macrophages Pre-Exposed to Amorphous Silica Nanoparticles

**DOI:** 10.3390/nano12132307

**Published:** 2022-07-05

**Authors:** Massimiliano G. Bianchi, Martina Chiu, Giuseppe Taurino, Enrico Bergamaschi, Francesco Cubadda, Guido M. Macaluso, Ovidio Bussolati

**Affiliations:** 1Department of Medicine and Surgery, University of Parma, 43126 Parma, Italy; martina.chiu@unipr.it (M.C.); giuseppe.taurino@unipr.it (G.T.); guidomaria.macaluso@unipr.it (G.M.M.); ovidio.bussolati@unipr.it (O.B.); 2Department of Public Health Sciences and Pediatrics, University of Turin, 10126 Turin, Italy; enrico.bergamaschi@unito.it; 3Istituto Superiore di Sanità-Italian National Institute of Health, 00161 Rome, Italy; francesco.cubadda@iss.it; 4Centro di Odontoiatria, University of Parma, 43126 Parma, Italy; 5Microbiome Research Hub, University of Parma, 43124 Parma, Italy

**Keywords:** autophagy, endotoxin, glutamine synthetase, inflammation, monocyte-derived macrophages, synthetic amorphous silica (SAS), toll-like receptor 4

## Abstract

Amorphous silica nanoparticles (ASNP) are present in a variety of products and their biological effects are actively investigated. Although several studies have documented pro-inflammatory effects of ASNP, the possibility that they also modify the response of innate immunity cells to natural activators has not been thoroughly investigated. Here, we study the effects of pyrogenic ASNP on the LPS-dependent activation of human macrophages differentiated from peripheral blood monocytes. In macrophages, 24 h of pre-exposure to non-cytotoxic doses of ASNP markedly inhibited the LPS-dependent induction of pro-inflammatory (TNFα, IL-6) and anti-inflammatory cytokines (IL-10). The inhibitory effect was associated with the suppression of NFκB activation and the increased intracellular sequestration of the TLR4 receptor. The late induction of glutamine synthetase (GS) by LPS was also prevented by pre-exposure to ASNP, while GS silencing did not interfere with cytokine secretion. It is concluded that (i) macrophages exposed to ASNP are less sensitive to LPS-dependent activation and (ii) GS induction by LPS is likely secondary to the stimulation of cytokine secretion. The observed interference with LPS effects may point to a dampening of the acute inflammatory response after exposure to ASNP in humans.

## 1. Introduction

Amorphous silica nanoparticles (ASNP) have been the subject of an increasing number of toxicological studies given that they are present in a variety of common consumer products and widely used for biomedical applications and as a food additive [1,2,3,4,5]. Although synthetic amorphous silica has been considered safe for many decades, recent investigations have highlighted the pro-inflammatory effects of ASNPs [6,7,8,9,10,11,12,13] as well as their hazard potential and accumulation in several organs [14,15]. In the growing debate about the biological effects of engineered nanomaterials (ENM), an issue of interest consists of the possibility that they interfere with the response of immune cells to other agents, potentially affecting inflammation pathways [16]. For example, ASNPs potentiate activation by LPS, at least in terms of NO production, of murine macrophage Raw 264.7 cells [17]. More recently, we have shown that exposure to low doses of ASNP interferes with the induction of glutamine synthetase, a change potentially related to macrophage polarization [18], and delays IL-1β secretion in LPS-treated THP-1 human macrophage-like cells [19]. In both cases, pyrogenic NM-203 ASNPs produced larger effects than the precipitated NM-200 ASNPs, confirming their higher bioreactivity observed in studies with other cell models [17,19]. These data suggest that the interaction of ASNP with innate immune cells may have more complex effects than pro-inflammatory activation, such as alterations in the time course and features of the cell response to LPS.

Although being of human origin and widely used in nanotoxicological studies on ASNP [19,20,21,22,23,24,25], THP-1 macrophages have some limitations when fine biological mechanisms underlying the inflammatory response are to be investigated. Indeed, besides being a leukemic cell line, THP-1 attain a macrophage-like phenotype and pro-inflammatory competences only if differentiated with phorbol esters such as phorbol myristate acetate (PMA). However, in a variety of cell models, phorbol esters, producing a chronic activation of several PKC isozymes, modify NFκB-dependent and other transduction pathways [26,27,28] and, therefore, may confound the effects of ASNP.

For this reason, in the present study we have characterized the effects of exposure to ASNP on LPS-dependent activation of primary human macrophages derived from peripheral blood monocytes (HMDM). Moreover, to mimic real-life conditions more closely, we cultured cells with human serum and exposed them to low, nontoxic doses of the pyrogenic NM-203 ASNP before LPS stimulation.

## 2. Materials and Methods

### 2.1. Materials

Pyrogenic amorphous silica nanoparticles (ASNP) NM-203 were from the same batch used in [19] and were obtained from the JRC Nanomaterials Repository (Ispra, Varese, Italy). The physico-chemical characterization of NM-203 ASNPs, before and after their suspension in different media, has been extensively reported elsewhere [17]. A/B Human serum and *E. coli* LPS (strain O55:B5) were from Sigma-Aldrich (Merck, Milan, Italy), while RPMI 1640 culture medium was provided by Corning (New York, NY, USA). The antibodies used were mouse monoclonal anti-GS (BD, Franklin Lakes, NJ, USA), rabbit polyclonal anti-LC3BI/II, rabbit polyclonal anti-p-NFκB, mouse monoclonal anti-NFκB, mouse monoclonal anti-LAMP1 (Cell Signaling, Danvers, MA, USA), rabbit polyclonal anti-TLR4 (Novus Biologicals, Cambridge, UK). All antibodies were diluted 1:1000 unless otherwise specified.

### 2.2. Monocyte Isolation and Macrophage Differentiation

Peripheral blood mononuclear cells (PBMC) were isolated starting from peripheral blood aliquots obtained from healthy donors upon approval of the local institutional review committee (# 3182/2018).

Aliquots of 30 mL of blood were diluted 1:3 in phosphate buffered saline (PBS), and equal volumes of the solution (22 mL) were carefully dispensed on 15 mL of Ficoll-Paque™ Plus (GE Healthcare Life Sciences, Pittsburgh, PA, USA) in four sterile conical tubes. After 30 min of centrifugation at 400× *g* in a swinging-bucket rotor without brake, buffy-coat fractions of PBMC were harvested and pulled into a new tube. Cells were then suspended, pelleted at 200× *g* for 10 min and repeatedly washed with 50 mL of PBS to remove red blood cells and platelet contamination. Purified PBMCs were resuspended in RPMI1640 culture medium supplemented with 5% of A/B human serum and seeded either on 24-well culture plate, to perform Western blot and RT-PCR experiments, or on 4-well chamber slides for immunofluorescence (Falcon). For all the experiments, a volume of 1 mL of cell suspension, corresponding to 3 × 10^5^ cell, was dispensed into each well. After 30 min of incubation in standard culture conditions (37 °C at 5% CO_2_), adherent monocytes were rinsed three times with PBS to remove residual lymphocytes and platelets. Culture medium was then renewed, and cells were incubated for 7 days in the presence of M-CSF (R&D System, 50 ng/mL) to achieve macrophage differentiation. Cells were then kept in culture for further 24 h in the absence of M-CSF before exposure to ASNP.

### 2.3. ASNP Suspension and Experimental Treatments

To remove endotoxin contamination, ASNPs were heated at 230 °C for 4 h [17]. Afterwards, they were suspended at a concentration of 12.8 mg/mL in pyrogen-free water with 0.005% Bovine Serum Albumin (BSA), sonicated for 16 min (from Nanogenotox protocol, with modifications [17,19]), and diluted in complete culture medium at different working concentrations. For cell viability and cytotoxicity assays, cultured cells were exposed from 2.5 up to 10 µg/cm^2^ of ASNP for 48 h, while for all the other experiments MDMs were pre-exposed to 10 µg/cm^2^ of ASNP for 24 h and then incubated in the presence or in the absence of LPS (1 ng/mL) for the last 24 h. For the experimental treatments, cells were differentially processed depending on the biological endpoint, as described for each experiment.

### 2.4. Cell Viability and Cytotoxicity

To assess ASNPs’ toxicity and their impact on cell viability, macrophages were differentiated in 96-well culture plates and treated as described above. Cytotoxicity was determined with the Lactate Dehydrogenase Activity Assay Kit (Merck), following the procedure suggested by the manufacturer, while cell viability was analyzed with the resazurin method, as described in [19]. Protein determination was performed with the Lowry method.

### 2.5. Western Blot Analysis

After the experimental treatments, culture medium was harvested and stored at −80 °C to be processed for ELISA assay (see below). Adherent macrophages were then rinsed once in PBS and covered with 80 µL of Laemmli sample buffer (SB) 4× (250 mM Tris–HCl, pH 6.8, 8% SDS, 40% glycerol, and 0.4 M DTT) previously diluted at the working concentration in lysis buffer (20 mM Tris–HCl, pH 7.5, 150 mM NaCl, 1 mM EDTA, 1 mM EGTA, 1% Triton, 2.5 mM sodium pyrophosphate, 1 mM β-glycerophosphate, 1 mM Na_3_VO_4_, 1 mM NaF, and 2 mM imidazole), supplemented with a protease inhibitor cocktail (Complete Mini, EDTA-free, Roche, Monza, Italy). A pool of extracts from two wells was prepared for each experimental condition. Cell lysates were boiled for 10 min and protein concentration was determined with the Lowry method. Volumes of lysate corresponding to 20 µg of proteins were loaded on 10% SDS-polyacrylamide gels, separated at 100 V for 1.5 h and then transferred overnight at 4 °C on PVDF membranes. Membranes were rinsed once with Tris-buffered saline (TBS; 50 mM Tris Base, 150 mM NaCl, pH 7.5) and incubated for 1 h in TBS 10% of blocking solution (Western Blocking Reagent, Roche) before being exposed overnight at 4 °C to primary antibodies diluted in TBS-T (TBS with 0.1% Tween) containing 5% BSA. Membranes were then rinsed three times in TBS-T and incubated for 1 h at RT in HRP-conjugated secondary anti-mouse or anti-rabbit antibodies, both diluted 1:10,000 in blocking solution. After that, membranes underwent three washes of 15 min each with TBS-T before being exposed to ECL Chemiluminescent HRP Substrate (Millipore, Merck, Milan, Italy) for 1 min. Visualization of the protein bands was performed using the iBright™ FL1500 automated system (Life Technology, Thermo Fisher, Milan, Italy). The expression of β-actin was used for loading control.

### 2.6. Immunocytochemistry and Confocal Microscopy

For cell staining, macrophages were rinsed twice in PBS and fixed for 15 min with 4% paraformaldehyde in PBS. After two washes in PBS, fixed cells were permeabilized 15 min with 0.2% Triton in PBS, blocked for 1 h in PBS 2% BSA plus 2% normal goat serum (NGS) and incubated overnight at 4 °C with mouse anti-GS (1:100), rabbit anti-TLR4 (1:200) or mouse anti-LAMP1 (1:150) primary antibodies diluted in blocking solution. After three washes in PBS, cells were incubated 1 h in the presence of AlexaFluor 568 anti-mouse for GS, AlexaFluor 546 anti-rabbit for TLR4 and AlexaFluor 488 anti-mouse for LAMP1 all diluted 1:400 in blocking solution. In cells stained for GS analysis, actin cytoskeleton was visualized adding AlexaFluor 633 phalloidin (3 U/mL) in the last 15 min of incubation. For nuclei counterstaining, 1 µg/mL of Hoechst 33,258 was added to all the conditions tested. Slides were then mounted with a coverslide in Glycergel medium (DAKO, Santa Clara, CA, USA) and observed with the CLSM system Stellaris 5 (Leica Microsystems, Wetzlar, Germany) using a Plan-Apo 63× oil immersion objective (NA 1.4). Single confocal sections were taken adopting independent configuration settings to avoid any possible crosstalk during acquisition. For the analysis of GS, following configuration was used: setting 1, excitation at UV 405 nm LL (Laser Line) and 579 nm WLL (White Laser Line) with a double spectral detection ranging from 420 to 468 nm and 583 to 621 nm for the acquisition of the signals corresponding to nuclei and GS, respectively; setting 2, excitation at 631 nm WLL and a spectral detection range from 639 to 695 nm for the acquisition corresponding to actin. For the study of TLR4 and LAMP1 distribution, the acquisition settings were, respectively, excitation at 549 nm WLL with a spectral detection range of 570–610 nm and excitation at 488 nm WLL with a spectral detection range of 500–530 nm.

### 2.7. Gene Silencing

GS silencing was performed as reported in [29] with some modifications. Macrophages were cultured for 24 h in complete culture medium and then transfected with scramble siRNA (ON-TARGETplus Non-targeting Pool, DharmaconTM, Lafayette, CO, USA) or with siRNAs targeting GLUL (ON-TARGETplus Human GLUL siRNA-SMARTpool, DharmaconTM), following manufacturer’s procedure. Briefly, cells were washed in PBS and incubated in serum- and antibiotic-free RPMI supplemented with 6 µL/mL of DharmaFECT (DharmaconTM) transfection reagent and siRNAs (25 nM). After 18 h, medium was replaced with fresh standard growth medium. Kept in culture for further 24 h, GLUL-silenced or scrambled-transfected cells were treated with LPS (1 ng/mL) and processed for either Western blot or RT-PCR analysis. In parallel, aliquots of culture media were conserved at –80 °C for ELISA assay.

### 2.8. RNA Extraction and Real Time PCR

The extraction of total RNA was performed using the GeneJET RNA Purification Kit (Life Technology, Thermo Fisher), following manufacturer’s instructions. In brief, 250 ng of total RNA were reverse-transcribed and aliquots of 12 ng of cDNA were amplified in a total volume of 10 μL with the Power Up SYBR Green Master mix (ThermoScientific, Milan, Italy), along with the forward and reverse primers (5 pmol each) reported in Table 1. Real-time PCR was performed in a Step One Plus apparatus (Life Technology, Thermo Fisher) setting a denaturation step at 95 °C for 15 s, which was followed by the annealing (15 s) and extension (1 min) steps at a temperature characteristic for each pair of primers (Table 1). Fluorescence was monitored at the end of each extension. Melting curve analysis was added at the end of each amplification cycle. Data analysis was made according to the relative standard curve method. Expression data were reported as the ratio between each investigated mRNA and RPL15 mRNA.

### 2.9. Statistics

Data were analyzed with Prism 6.1 (GraphPad, La Jolla, CA, USA). Values are reported as means ± SD. Differences between the groups were evaluated with a *t*-test or one-way ANOVA, followed by appropriate multiple comparison tests when significant, as indicated in the figure legends. Differences were considered significant when *p* < 0.05.

## 3. Results

### 3.1. The Exposure to ASNP Did Not Cause Cytotoxicity in Human Macrophages

To assess the cytotoxic effects of ASNP exposure in primary cultures of human monocyte-derived macrophages (MDM), we evaluated cell viability after 48 h of exposure to increasing doses of pyrogenic ASNP NM-203 (Figure 1A), either in the absence or in the presence of 1 ng/mL LPS. ASNP did not affect substantially cell viability or protein content of the cell culture, either in the absence or in the presence of LPS (Figure 1A,B). At the highest dose used (10 µg/cm^2^), only a slight decrease in viability was observed in the presence of LPS, while no significant cytotoxicity was detected both in the presence and in the absence of the endotoxin (Figure 1C). This dose was adopted in all the following experimental approaches.

### 3.2. ASNPs Dampen the Secretion of Inflammatory Cytokines

To investigate the impact of ASNP exposure on the inflammatory response of innate immune cells, we measured the secretion of TNF-α, IL-6 and IL-10 in culture medium of macrophages exposed to ASNP for 48 h and challenged with LPS in the last 24 h. As expected, endotoxin caused a marked secretion of the cytokines (Figure 2A). ASNP exposure alone slightly increased only IL-10 secretion, but significantly inhibited the LPS-dependent secretion of all the cytokines (Figure 2A). As shown in Figure 2B, this effect was associated with lack of stimulation of cytokine gene expression by 2 h treatment with LPS in ASNP-pre-treated macrophages.

### 3.3. ASNP Inhibit NFκB Activation, Affect Autophagic Flux and Promote TLR4 Internalization

Pro-inflammatory cytokine induction by LPS in macrophages depends on the binding of the endotoxin to the TLR4 receptor and the consequent activation of the NFκB pathway. To clarify the mechanism underlying the ASNP-dependent inhibition of cytokines production upon endotoxin challenge, we investigated the effect of LPS treatment on the activation of the transcription factor p65-NFκB in control or ASNP-exposed macrophages. As expected, a 30 min incubation with LPS increased the abundance of the phosphorylated form of NFκB, an effect that was lowered by ASNP exposure (Figure 3A).

While neither LPS nor ASNP affected the overall protein abundance of the LPS receptor TLR4 (Figure 3B), 48 h of ASNP exposure led to changes in its localization. In a portion of cells, TLR4 was internalized and exhibited a partial co-localization with the lysosomal marker LAMP1, particularly evident in LPS-treated cells pre-exposed to ASNP (Figure 3C, Appendix A). We then investigated the effect of ASNP on the autophagic flux in MDM-exposed cells (Figure 4). Silica NP significantly increased LC3BII abundance without affecting that of SQSTM1/p62 protein. On the contrary, LPS treatment did not modify the expression of the two autophagic markers, either in ASNP-exposed or not exposed cells.

### 3.4. LPS-Dependent Induction of Glutamine Synthetase Gene and Protein Expression Are Prevented by ASNP Exposure

As previously reported in the human macrophage cell line THP-1 [19], the treatment with LPS of human MDM caused a time-dependent increase in glutamine synthetase (GS) expression, an effect that was maximal after 24 h of incubation (Figure 5A) and attributable to a sizable induction of the *GLUL* gene (Figure 5B). While ASNP alone did not modify GS expression levels detected in cells not treated with endotoxin, pre-exposure to ASNP suppressed the LPS-dependent GS induction at both gene and protein levels (Figure 5C,D). These data were consistent with the observations performed with confocal microscopy (Figure 6). The images reveal a cytoplasmic distribution of GS signal, also showing some protein clusters mainly localized in the perinuclear region. In LPS-treated macrophages, a significant increase in GS signal was detected, an effect that was hindered in cells pre-exposed to ASNP, without evident changes in enzyme distribution (Figure 6, bottom). In the same experimental condition, we also investigated the expression of ASCT2 protein, a Na^+^-dependent transporter system for amino acids, which accounts for most of Gln uptake across the plasma membrane of several mammalian cells. We found that neither ASNP nor LPS affected the high expression levels of ASCT2 in MDM cells (data not shown).

### 3.5. GS Silencing Does Not Affect Cytokine Secretion by LPS-Stimulated Macrophages

To explore the potential role of glutamine synthetase on LPS- and ASNP-dependent changes in the secretion of inflammatory cytokines by human MDM, we silenced *GLUL* expression and then treated cells with LPS for 24 h. The results, reported in Figure 6, showed that the targeting of *GLUL* significantly lowered the basal levels of GS and abolished the LPS-dependent induction of the enzyme at both the gene and protein level (Figure 7A,B). However, GS silencing did not prevent the secretion of TNF-α or IL-10 triggered by LPS (Figure 7C,D), suggesting that the production of the two cytokines did not require GS in human macrophages.

## 4. Discussion

The results obtained in the present contribution demonstrate that exposure to pyrogenic ASNP modifies and hampers the acute, pro-inflammatory activation of primary human macrophages. This effect is observed with a dose of ASNP not causing significant cytotoxicity. While the focus of the study is the characterization of the immunomodulatory properties of SAS as a nanomaterial, the absence of cytotoxicity is of pivotal importance, since the release of intracellular contents able to behave as DAMPs (damage-associated molecular patterns) and, hence, activate macrophage receptors, may obviously have pro-inflammatory effects. However, also nanomaterials at non-toxic exposure doses may influence the progression of inflammatory responses, a particularly important circumstance for evaluating biological consequences of real-life exposure to nanomaterials [16].

Non-toxic doses of ASNP affect the response to LPS of THP-1 macrophage-like cells [19], blocking autophagic flux, preventing the LPS-dependent induction of glutamine synthetase expression and, more importantly, delaying LPS-dependent IL-1β secretion. These results suggest a negative modulation of inflammatory activation by the nanomaterial. While ASNPs are usually considered pro-inflammatory when used on naïve innate immune cells, the possibility that they interfere with macrophage activation by pathogen-associated molecular patterns (PAMPs) has not been thoroughly explored. However, this hypothesis is strongly supported by the results presented here. While ASNPs alone did not markedly modify gene induction or the amounts of secreted cytokines, they significantly blunted LPS-dependent stimulation of cytokine production. In terms of secretion, the inhibitions observed ranged from over 60% for TNFα to almost 80% for IL-6. However, even in the presence of ASNP, LPS still significantly stimulated cytokine secretion, with the only exception of IL-10, the levels of which, in cells exposed to the nanomaterial, were not markedly different in the absence or in the presence of LPS.

The sequence of events that lead to LPS-dependent cytokine induction in macrophages is well understood, starting from the interaction of the endotoxin with TLR4 and triggering the consequent activation of NFκB through the MyD88 adaptor protein. NFκB activation by LPS is almost suppressed by ASNP in MDM (Figure 3A). Conversely, NFκB activation was not apparently affected by ASNP in THP-1 cells, and the expression of genes induced by this pathway was, consistently, not affected by the nanomaterial [19]. This apparent incongruence may depend on the NFκB activation caused by the PMA treatment needed to obtain THP-1 differentiation to macrophages [26,27,28]. On the other hand, the LPS concentration used to activate THP-1 cells in [19] was 100 ng/mL, 100-fold higher than the one used here, a condition that likely caused a larger NFκB activation, no more inhibitable by ASNP.

As far as TLR4 is concerned, no substantial modulation of its expression by LPS or ASNP is detected under the conditions tested. A detailed analysis of receptor distribution among the different cell compartments would require more quantitative techniques such as biotinylation that could be uneasy given the not-high expression of TLR4 in M0 MDM. However, the presented results on receptor distribution (Figure 3C) indicate a clear cut intracellular clusterization of TLR4 in ASNP-treated cells, both in the absence and, even more evidently, in the presence of LPS, when a partial co-localization of the receptor with the lysosomal marker LAMP1 is also evident. Although autophagy is stimulated by ASNP in MDM, as indicated by the increased abundance of the marker LC3BII (Figure 4), SQSTM1 (p62) expression is not changed, suggesting an expansion of the autophagic compartment, with possible entrapment of TLR4, rather than an acceleration of the autophagic flux. Similar effects of ASNP have been described in another cell model [30]. It is known that LPS causes the arrest of MyD88-dependent signal transduction and NFκB activation [31]. It is therefore possible that ASNPs enhance TLR4 receptor internalization and promote their trapping, lowering the sensitivity of macrophages to LPS and blunting NFκB activation. A working model of the proposed mechanism is shown in the graphical abstract. It will be interesting to study if ASNP-dependent enhanced internalization of TLR4 also causes a shift in the transduction pathways activated by the endotoxin, as described for TiO_2_ nanoparticles in LPS-treated Raw 264.7 murine macrophages [32]. In particular, the possible involvement of MAPKs should be specifically investigated.

As much as we know, it is the first time that ASNP-induced internalization of TLR4 is described in primary human M0 macrophages. It is known that in different cells, silica NP internalization might involve several mechanisms such as, for example, caveolin-initiated macropinocytosis in alveolar epithelial cells [33]; LDLR-dependent internalization, which does not involve a clathrin-mediated mechanism and depends on the protein-corona adsorbed on silica NP [34]; and, more interestingly, the recent contribution by Wang et al. [35] where surfactant-associated proteins A and D (SP-A/D) opsonize ENMs enhancing their phagocytosis by alveolar macrophages through a mechanism that requires the TLR4 coreceptor CD14. Thus, it is possible that also NM203, the ASNP used in our study, interacts with CD14 causing the internalization of the TLR4-CD14 complex. However, the possibility of CD14 involvement in the changes reported here will require experimental validation.

LPS stimulates GS expression through *GLUL* induction, suggesting changes in the amino acid metabolism associated with endotoxin-dependent activation. The effect on GS is specific since the Gln transporter ASCT2 does not increase upon LPS stimulation (not shown). As in THP-1 cells [19], ASNPs prevent the LPS-dependent increase in GS expression, attributable to the induction of the *GLUL* gene also in MDM. ASNPs do not change the basal expression of GS (Figure 5C,D), or its distribution (Figure 6) compared to control cells. However, even if GS silencing markedly lowers LPS-dependent *GLUL* induction (Figure 7A,B), it does not modify LPS-stimulated cytokine secretion, at least in the short experimental times adopted here (Figure 7C,D). Thus, it is unlikely that ASNP modulation of cytokine secretion is linked to ASNP effects on GS expression. In LPS-stimulated microglia [36] and in M2-polarized macrophages [37], GS inhibition stimulates the secretion of pro-inflammatory cytokines and a M1-shift, suggesting that the enzyme plays a regulatory role in pro-inflammatory activation. Recent results in vivo seem to confirm this hypothesis [38]. The blockade of the LPS-dependent induction of GS expression, observed in THP-1 cells and here confirmed in MDM pre-exposed to ASNP, may have functional consequences affecting M1-M2 phenotypic skewing. Indeed, if GS induction represents a determinant of the M2 phenotype (see refs. [36,37]), ASNP exposure would prevent this polarization. Long-term experiments on marker characterization will clarify this interesting issue. This hypothesis would represent another modulatory mechanism by which ASNP could divert the outcome of acute inflammation.

## 5. Conclusions

In conclusion, these results suggest a mechanistic hypothesis in which ASNPs lead to the intracellular sequestration of a significant fraction of TLR4, rendering macrophages less sensitive to LPS stimulation, hampering NFκB activation and substantially lowering the induction of genes dependent on this pathway (see graphical abstract). From a functional point of view, the impairment of inflammatory response may have short-term consequences, such as a reduced efficiency of acute responses to PAMPs. However, in the long term, sub-optimal responses may also hinder resolution, thus favoring evolution towards chronicity.

## Figures and Tables

**Figure 1 nanomaterials-12-02307-f001:**
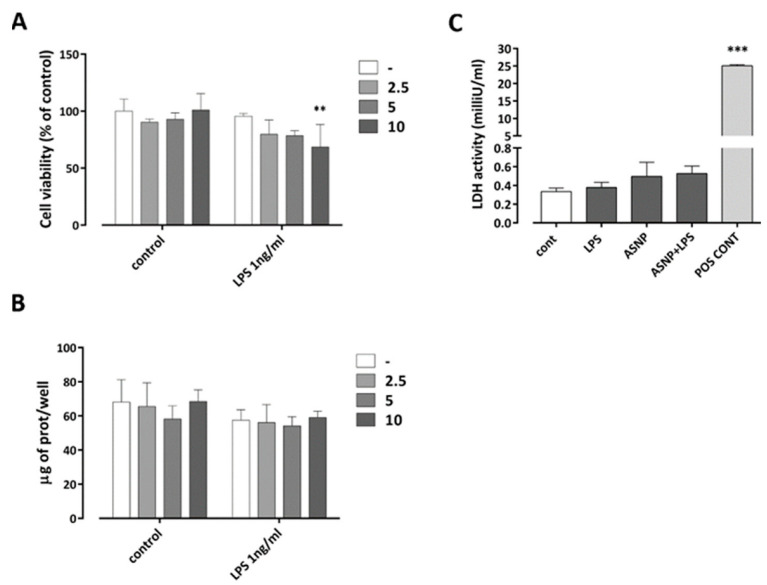
**ASNP toxicity on human macrophages.** Monocytes were isolated and differentiated as described in Materials and Methods. Macrophages were then exposed to increasing doses (2.5 up to 10 µg/cm^2^; (**A**,**B**)) or to 10 µg/cm^2^ (**C**) of ASNP for 48 h. In the last 24 h, incubation was performed in the presence or in the absence of LPS (1 ng/mL). After the experimental treatments, cell viability (**A**) and protein content (**B**) were determined. Culture medium was conserved for the LDH activity assay (**C**). Data of cell viability and protein abundance are means of three independent determinations ± SD in two experiments, while LDH values are means of six independent determinations ± SD in two experiments. Positive control (**C**); medium of cells incubated in the presence of Triton-X100 0.01% for 15 min. ** *p* < 0.01 vs. LPS 1 ng/mL without ASNP; *** *p* < 0.001 vs. control, two-way ANOVA. In (**A**,**C**), Sidak multiple comparison post hoc test was performed.

**Figure 2 nanomaterials-12-02307-f002:**
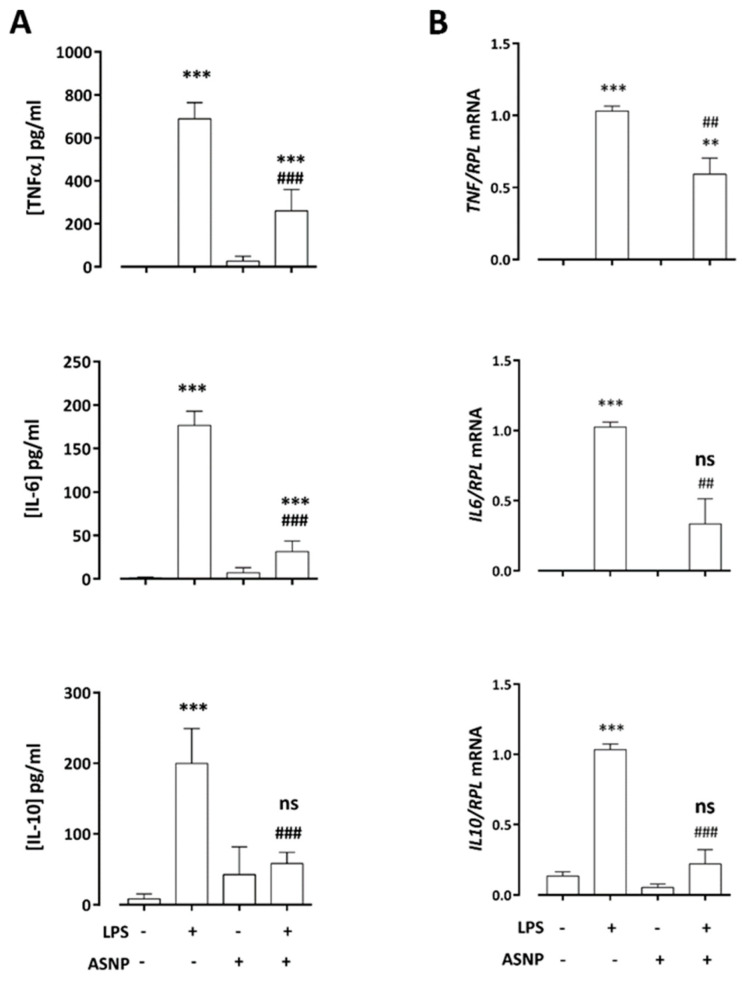
**Effects of ASNP exposure on LPS-dependent cytokine induction.** Macrophages were exposed to ASNP (10 µg/cm^2^), as described in Materials and Methods, and challenged with LPS 1 ng/mL in the last 24 h of incubation. (**A**) Secreted cytokines after 24 h of LPS treatment. Data are means of six determinations ± SD obtained from two independent experiments. (**B**) Gene expression after 2 h of LPS treatment. Data are means of four determinations ± SD in two independent experiments. ** *p* < 0.01, *** *p* < 0.001 vs. control; ## *p* < 0.01, ### *p* < 0.001 vs. LPS 1 ng/mL; ns = not significant vs. control. (**A**), one-way ANOVA followed by a Tukey’s multiple comparisons test; (**B**), two-tail *t*-test for unpaired data.

**Figure 3 nanomaterials-12-02307-f003:**
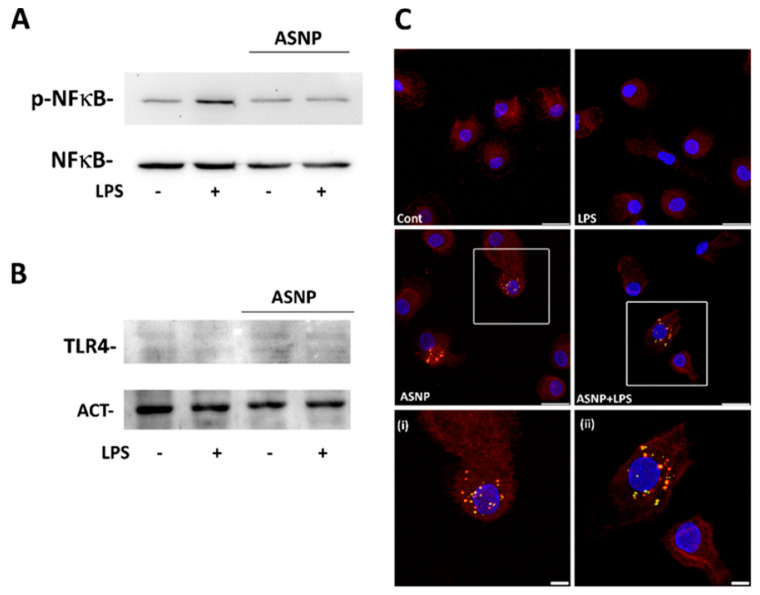
**ASNP impact on LPS−dependent NFκB activation and TLR4 distribution.** Macrophages were exposed to ASNP, as described above, and then incubated in the presence or in the absence of LPS 1 ng/mL for 30 min (**A**) or 24 h (**B**,**C**). After the experimental treatment, cells were processed for Western blot or immunocytochemistry. (**A**,**B**), Two representative Western blots performed twice with comparable results. (**C**), Maximal projection of single confocal sections of representative fields of MDM cultures stained for TLR4 (red) and LAMP1 (green). Nuclei are counterstained in blue (Bars = 20 µm); (**i**) and (**ii**), magnifications of the regions of interest highlighted in the whole fields (Bars = 5 µm).

**Figure 4 nanomaterials-12-02307-f004:**
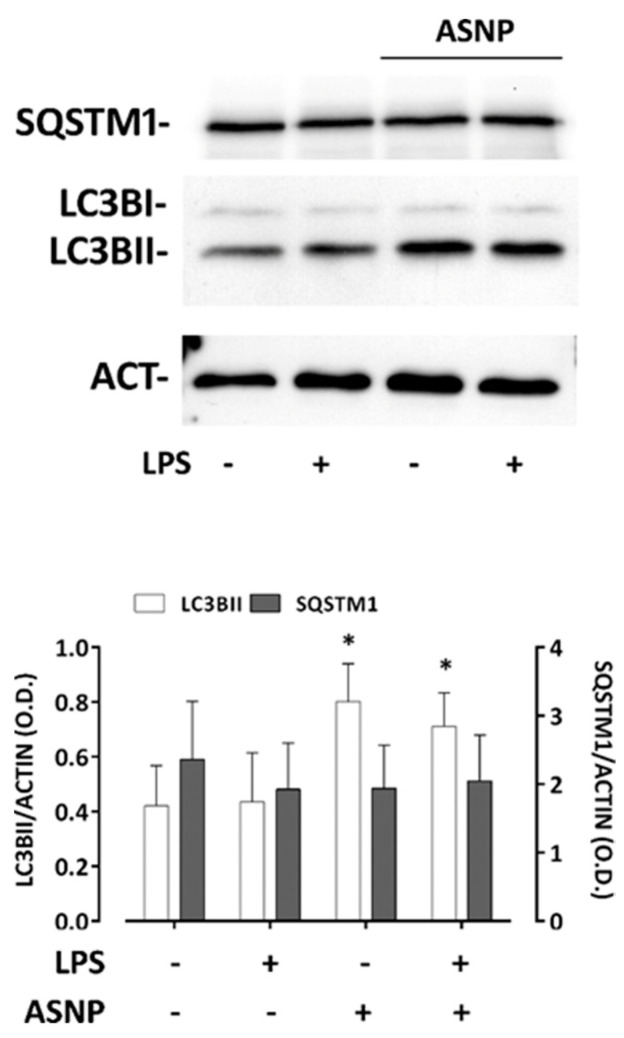
**Effect of ASNP exposure on autophagic flux.** Macrophages were treated as described above (refer to Figure 2) and then processed for Western blot analysis. Top, a representative Western blot, showing the expression of LC3BI/II and SQSTM1, is reported. ACT, β−actin for loading control. Bottom, data of densitometric analysis. Data are means ± SD of the intensity bands of the two proteins obtained in three independent experiments. * *p* < 0.05 vs. control, two-tail t-test for unpaired data.

**Figure 5 nanomaterials-12-02307-f005:**
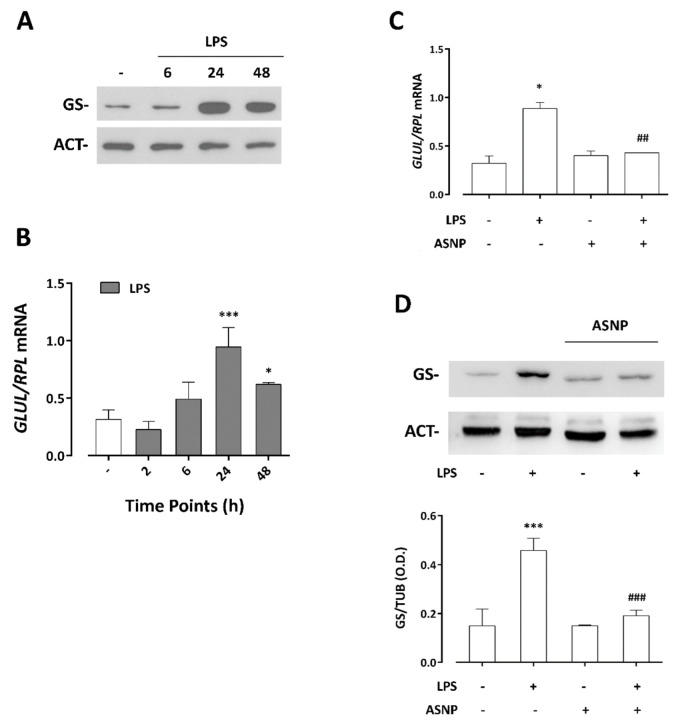
**Effect of LPS treatment on GS expression in human macrophages exposed to ASNP.** (**A**,**B**). Macrophages were treated with LPS for the times indicated and then processed for either Western blot or RT−PCR analysis. (**A**), A representative Western blot performed twice with comparable results. (**B**), The relative expression of *GLUL* mRNA. Data are means of four determinations ± SD obtained from two independent experiments. (**C**,**D**), Cells were treated as described for Figure 2 and then processed for RT−PCR or Western blot. (**C**), The relative expression of *GLUL* mRNA. Data are means of four determinations ± SD obtained from two independent experiments. (**D**), A representative Western blot along with the relative abundance of GS protein is reported. Data are the means of three independent experiments ± SD. For all the panels, * *p* < 0.05, *** *p* < 0.001 vs. control; ## *p* < 0.01, ### *p* < 0.001 vs. 1 ng/mL LPS. (**B**,**D**), One−way ANOVA followed by Tukey’s multiple comparison test; (**C**), two−tail *t*−test for unpaired data.

**Figure 6 nanomaterials-12-02307-f006:**
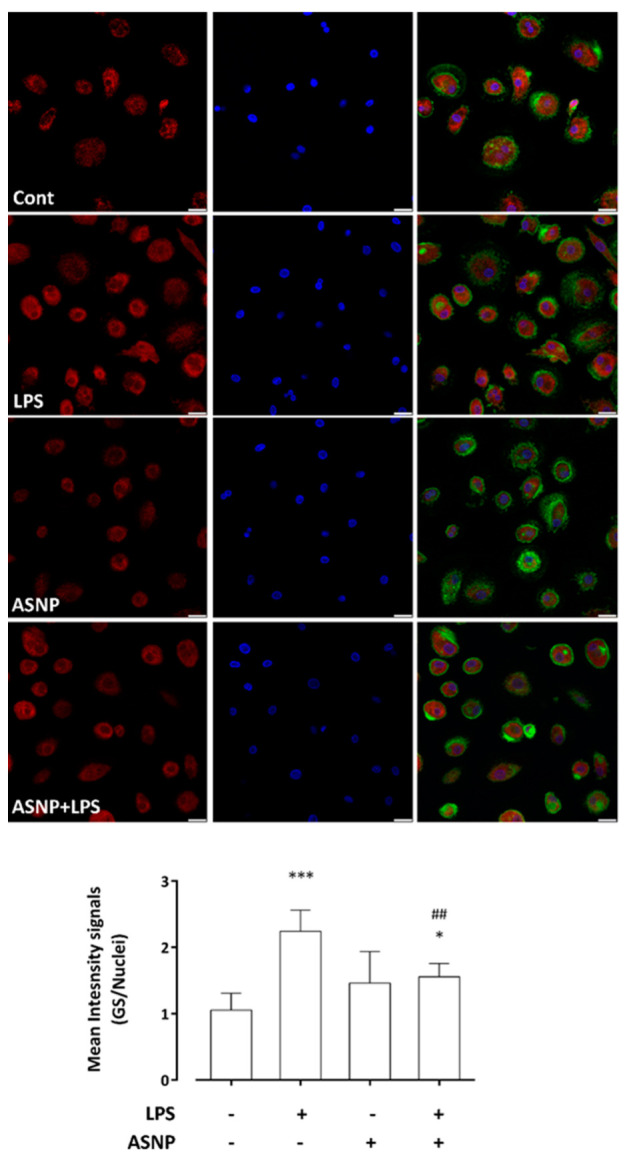
**GS protein expression in human MDM exposed to ASNP.** After the experimental treatments (ASNP, 48 h; LPS, the last 24 h of incubation), human macrophages were fixed and co−stained for GS (red) or actin (green), as described in Materials and Methods. Nuclei counterstaining is reported in blue. Images are representative fields of confocal single sections. Bars = 20 μm. The results are the mean ± SD of GS/nuclei intensity signals, recorded on the acquired single sections (n = 8). * *p* < 0.05, *** *p* < 0.001 vs. Control; ## *p* < 0.01 vs. 1 ng/mL LPS, one−way ANOVA followed by Tukey’s multiple comparison test.

**Figure 7 nanomaterials-12-02307-f007:**
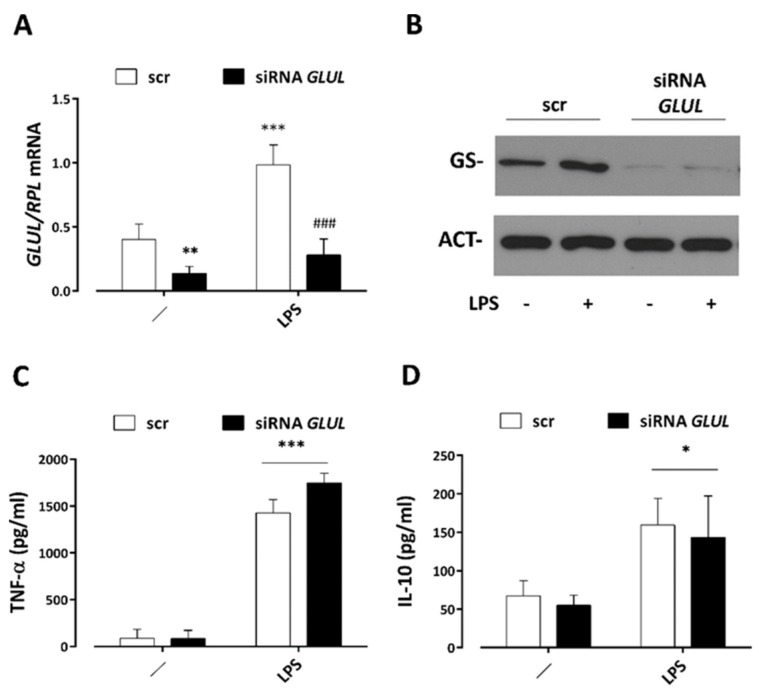
**GS silencing and cytokine secretion in LPS**−**stimulated macrophages.** Macrophages were transfected with scr or *GLUL*−siRNA, as described in Materials and Methods. Cultured cells were then challenged for 24 h with 1 ng/mL LPS before being processed for RT−qPCR (**A**) or Western blot (**B**). In parallel, culture medium was conserved for the determination of secreted cytokines (**C**,**D**). (**A**), Data are means ± SD of four determinations in two independent experiments, ** *p* < 0.01 vs. scr; ### *p* < 0.05 vs. scr in the presence of LPS, two−tail *t*−test for unpaired data. (**B**), A representative Western blot performed twice with comparable results. (**C**,**D**), Data are means ± SD of four independent determinations in two independent experiments, * *p* < 0.05, *** *p* < 0.001 vs. scr, one-way ANOVA followed by Tukey’s multiple comparisons test.

**Table 1 nanomaterials-12-02307-t001:** Primers used for studies of gene expression.

Gene	Forward	Reverse	T (°C)	Amplicon Size (bp)
Glutamine Synthetase (GLUL)	5′ TCA TCT TGC ATC GTG TGT GTG 3′	5′ CTT CAG ACC ATT CTC CTC CGG 3′	57 °C	137
Tumor necrosis factor alpha (TNF)	5’ ATG AGC ACT GAA AGC ATG ATC C 3’	5’ GAG GGC TGA TTA GAG AGA GGT C 3’	61 °C	196
Interleukin-6 (IL6)	5’ AAC CTG AAC CTT CCA AAG ATG G 3’	5’ TCT GGC TTG TTC CTC ACT ACT 3’	54 °C	159
Interleukin-10 (IL10)	5’ TCA AGG CGC ATG TGA ACT CC 3’	5’ GAT GTC AAA CTC ACT CAT GGC T 3’	56 °C	176
Ribosomal Protein L15 (RPL15)	5’ GCA GCC ATC AGG TAA GCC AAG 3’	5’ AGC GGA CCC TCA GAA GAA AGC 3’	57 °C	100

## Data Availability

Not applicable.

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
