# Peer review of "The TLR4/NFκB-Dependent Inflammatory Response Activated by LPS Is Inhibited in Human Macrophages Pre-Exposed to Amorphous Silica Nanoparticles"

_nanomaterials, 2022, doi:10.3390/nano12132307_

Round 1

Reviewer 1 Report

This paper focuses anti-inflammation effect of amorphous silica nanoparticles (ASNP) on macrophage induced by LPS. ASNP inhibited the LPS-dependent induction of pro-inflammatory cytokines, TNF-α and IL-6, and anti-inflammatory cytokine, IL-10. In addition, ASNP suppressed NF-kB activation induced by LPS. This paper contains valuable information and deserves to be published after some clarifications listed below.

(1) NF-kB is translocated from cytosol to nucleus by LPS stimulation via TLR4, and act as a promoter of transcription. In this paper, ASNP suppressed NF-kB activation. However, how about the translocation of NF-kB from cytoplasm to nucleus was not indicated. How about translocation of NF-kB?  In addition, how about the activation of MAP kinase?     

(2) How does ASNP enhance internalization of TLR4 on macrophage? Maybe, the quantification of TLR4 on cell surface can be easily indicated. What is the correlation between NF-kB activation and TLR4-internalization?

Author Response

Reviewer 1

Q: This paper focuses anti-inflammation effect of amorphous silica nanoparticles (ASNP) on macrophage induced by LPS. ASNP inhibited the LPS-dependent induction of pro-inflammatory cytokines, TNF-α and IL-6, and anti-inflammatory cytokine, IL-10. In addition, ASNP suppressed NF-kB activation induced by LPS. This paper contains valuable information and deserves to be published after some clarifications listed below.

R: We thank the Reviewer for the positive evaluation of our results.

Q:(1) NF-kB is translocated from cytosol to nucleus by LPS stimulation via TLR4, and act as a promoter of transcription. In this paper, ASNP suppressed NF-kB activation. However, how about the translocation of NF-kB from cytoplasm to nucleus was not indicated. How about translocation of NF-kB? In addition, how about the activation of MAP kinase?

R: We have not investigated here the translocation of NF-kB because, though it could be actually an interesting confirmation of our hypothesis, it wouldn’t give us any further information on the mechanisms involved. Indeed, LPS-dependent NF-kB phosphorylation (Figure 3A) is upstream of translocation and should prevent transcriptional activation, as here confirmed by the inhibition of cytokine gene induction (Figure 2B). As far as the potential involvement MAP kinases, we agree that it could well be possible since our previous results actually demonstrated the modulation of LPS-dependent activation of these pathways by nanomaterials (Bianchi et al. Toxicology Res. 2015 and Front. Immunol. 2017). However, we are well aware that our model is a minimal explanation of the experimental results, amenable of successive integrations. Among these, the role of the various MAPK (ERK1/2, P38 and JNK) will be certainly included. The possible limitations of the proposed model are now recognized in the discussion (lines 575-576).

Q: (2) How does ASNP enhance internalization of TLR4 on macrophage? Maybe, the quantification of TLR4 on cell surface can be easily indicated. What is the correlation between NF-kB activation and TLR4-internalization?

R: The two issues raised by the Reviewer are absolutely pertinent. We should preliminarily stress intrinsic limitations of our cell model in addressing this issue. Indeed, while human primary MDM are a most physiological model, they can be obviously obtained in limited number. Moreover, we use MDM in M0 state in short term experiments when the expression of TLR4 is low. This fact precludes either the quantification of TLR4 signal with confocal microscopy (unless potentially confounding amplification techniques are exploited) or the standard techniques for the assessment of membrane protein expression i.e. biotinylation which would require much larger number of cells (Bianchi et al. Neurochem. Int., 2006). Lacking these data, it is difficult to address the second question. However, since the LPS effect is not completely suppressed by ASNP pre-exposure, while ASNP effect on NFkB is assessed at a single time point, it is likely that other pathways are involved (see above MAPK) or that NFkB activation is time-shifted compared to control. These arguments are now presented in the Discussion (lines 557-559 and 575-576).

Reviewer 2 Report

Please provide the insight link between the pathways the authors analyzed to ASNP-induced toxicity in macrophages. 

Author Response

Reviewer 2

Q: Please provide the insight link between the pathways the authors analyzed to ASNP-induced toxicity in macrophages.

R: The model (graphical abstract) provides what is in our opinion the simplest mechanistic links between ASNP exposure and hindrance of macrophage response. As stated in the conclusions, we think that ASNP aggregates may lead to TLR4 intracellular sequestration causing a partial desensitization of macrophages. At variance with other contributions in the field, cells are pre-exposed to ASNP before LPS challenging mimicking the in vivo situation of a subject exposed to ASNP undergoing an inflammatory reaction. Therefore, incubation with nanoparticles causes a distortion of trafficking between cell membrane and endosomal compartments, here highlighted by morphological evidence of TLR4 sequestration and autophagy activation. Interestingly, Wang et al. Front. Immunol. 2021, now cited in the manuscript as ref. 36), describe CD14-dependent nanomaterials phagocytosis leading to internalization. Since CD14 and TLR4 form a complex on MDM membrane, it is tentative to speculate that ASNP may interact with the complex leading to its internalization (see revised discussion, lines 576-588).

Reviewer 3 Report

This article studied the response of innate immunity cells to pyrogenic NM-203 amorphous silica nanoparticles and found that the LPS activated pro-inflammatory responses can be dampened by the ASNP. The possible reason is the ASNP lead to internalization of a significant fraction of TLR4, rendering macrophages less sensitive to LPS stimulation, hampering NFκB activation and substantially lowering the induction of related genes. The use of primary human macrophages derived from peripheral blood monocytes (HMDM) is one of important novelty in the study. However, the authors provide some poor images from western blotting and confocal microscope which are not representative and convincing to support their findings. I would not recommend the paper to be published until the authors improve the quality of the images and address my other concerns in the next revision.

Major:

1.      It is known that LPS causes TLR4 internalization, while in Figure 3C, the confocal image of the +LPS group is similar to the control. And in the same figure, the authors only showed single cell images for ASNP and ASNP+LPS group, which is not convincing. Please show images which have more cells with TLR4 internalization and LAMP1 co-localization. 

2.      In figure 3B, the western blotting image showed band contamination in the -LPS/+ASNP group and both in TLR4 and actin. Please change those images.

3.      In Figure 5A, unknown bands (half cut) were lied on the top of GS band, this is very strange. You can clearly see there is no any band like that in Figure 5D, which is also GS experiment. I will doubt that one of them was cut at wrong kDa.

4.      The authors heated ASNP at 230°C for 4h to remove endotoxin, while for ASNP itself, I think it would be better to demonstrate the morphology and stability of particles did not change after the process.

5.      How can ASNP lead to the intracellular sequestration of a significant fraction of TLR4? Please address your hypothesis in the article. Can the ASNP be taken up by the cells via clathrin-dependent endocytosis pathways (receptor-mediated pathways)? I think the way of NP uptake may somehow affect the TLR4 internalization.  

 Minor:

1.      Line 553 and 576 : no graphical abstract can be seen.

2.      As in THP-1 cells [19], also in MDM ASNP prevented the LPS-dependent increase of GS expression, attributable to the induction of the GLUL gene.  Grammar should be checked

3.      Line 567-569: What do you mean “missed skewing ?”, you said GS inhibition stimulates the secretion of pro-inflammatory cytokines and a M1-shift, so I think in this sentence you believed the MDM exposed to ASNP will not be differentiated into M1 type macrophages, but M2 type, right?

4.      Line 556: RAW 264.7; also in all place in the article.

5.     Line 106-108: while for all the other experiments MDM were pre-exposed to 10 μg/cm2 of ASNP and then incubated in the presence or in the absence of LPS (1 ng/ml) for the last 24 h. What is the exposure duration for the ASNP? 24 h?

Author Response

Reviewer 3

This article studied the response of innate immunity cells to pyrogenic NM-203 amorphous silica nanoparticles and found that the LPS activated pro-inflammatory responses can be dampened by the ASNP. The possible reason is the ASNP lead to internalization of a significant fraction of TLR4, rendering macrophages less sensitive to LPS stimulation, hampering NFκB activation and substantially lowering the induction of related genes. The use of primary human macrophages derived from peripheral blood monocytes (HMDM) is one of important novelty in the study. However, the authors provide some poor images from western blotting and confocal microscope which are not representative and convincing to support their findings. I would not recommend the paper to be published until the authors improve the quality of the images and address my other concerns in the next revision.

R: We appreciate the pertinent criticisms of the Reviewer and her/his acknowledgment of the importance of the primary human macrophages as the cell model exploited in the study.

Major:

  • It is known that LPS causes TLR4 internalization, while in Figure 3C, the confocal image of the +LPS group is similar to the control. And in the same figure, the authors only showed single cell images for ASNP and ASNP+LPS group, which is not convincing. Please show images which have more cells with TLR4 internalization and LAMP1 co-localization.

R: Very few data are available concerning the LPS-dependent trafficking of TLR4-adaptor complex in human primary macrophages. Indeed, the contributions published to date and focused on this topic, one of which was cited in the original version of the manuscript, are mostly carried out on transfected cell models such as HEK-293 cells, endowed with a very high expression of TLR4 receptor, or on murine macrophages. These approaches have been always preferred to others involving endogenously expressing cell models, as the M0 macrophages used here, due to the relatively low expression levels of the TLR4 protein in normal/wild type cells. We think that the low expression can well explain why LPS-dependent TLR4 internalization is not very evident in cells not pre-exposed to ASNP. However, we have replaced control field with one in which a marginalization of a fraction of TLR4 is more evident in some of the cells. Moreover, in the whole figure 3C we have applied the maximal projection of the confocal single sections obtained through the Z-stack, as provided by the confocal software, so as to improve image rendering. The change is detailed in the figure 3 legend. We wish to stress, however, that difficulty in detecting LPS-dependent TLR4 internalization renders even more evident ASNP effect.

As the TLR4-LAMP1 co-localization is concerned, to cope with the Reviewer’s request we have added a Supplementary figure 1 showing single confocal sections of four distinct wide fields for each experimental condition, each or which include more cells than Figure 3C. In cultures pre-exposed to ASNP, we usually detect 2-3 cells, double positive for TLR4-LAMP1, with a clear-cut co-localization pattern, in a field of about 10 cells. To render clearer the message we have also changed the field representing the ASNP+LPS condition in Figure 3C. In all these fields both the internalization and the co-localization with LAMP1 are evident in a minority of cells pre-exposed to ASNP. We think that these cells are those that come into contact with larger quantities of ASNP agglomerates, since exposure of single cells to the nanomaterial is expected to be uneven due to sedimentation. However, although the phenomenon is evident in 30% of cells, the functional consequences are substantial given the high percentage of inhibition of cytokines secretion.

  1. In figure 3B, the western blotting image showed band contamination in the -LPS/+ASNP group and both in TLR4 and actin. Please change those images.

R: The image has been replaced with the image derived from another run of the same samples, as requested.

  1. In Figure 5A, unknown bands (half cut) were lied on the top of GS band, this is very strange. You can clearly see there is no any band like that in Figure 5D, which is also GS experiment. I will doubt that one of them was cut at wrong kDa.

R: We thank the Reviewer for the observation and apologize for the ambiguous message that the images could give. Actually, the band above GS (figure 5A of the original version) was not the result of a mismatch of the MW, but the revelation of another protein previously stained and incompletely stripped. This protein is the Gln transporter ASCT2, which does not change upon LPS stimulation or ASNP pre-exposure, as now stated as a result not shown in the text (lines 369-373, 589-591). This negative result, however, demonstrates that LPS-dependent GS changes are specific and do not involve all the enzyme/transporters implied in Gln metabolism. We have now performed a new run of the same samples and the results showing, as expected, GS band expression only are presented in the revised Figure 5A.

  1. The authors heated ASNP at 230°C for 4h to remove endotoxin, while for ASNP itself, I think it would be better to demonstrate the morphology and stability of particles did not change after the process.

R: The Reviewer is right in arising this doubt and to stress the importance of LPS elimination before studying effects on inflammation of nanomaterials, an often overlooked issue. However, the characterization of the ASNP preparation used here has been extensively described in the first study of our group in which this material was used (Di Cristo et al., Toxicol. Sci., 2016) and was performed after thermic treatment. Moreover, unpublished data on surface characteristics of the materials, such as zeta potential, obtained before and after thermic treatment, were comparable.

  1. How can ASNP lead to the intracellular sequestration of a significant fraction of TLR4? Please address your hypothesis in the article. Can the ASNP be taken up by the cells via clathrin-dependent endocytosis pathways (receptor-mediated pathways)? I think the way of NP uptake may somehow affect the TLR4 internalization.

R: We have not actually investigated the mechanism involved on ASNP/TLR4 internalization although it would be a very intriguing topic. As much as we know, it is the first time that this phenomenon is described in primary human M0 macrophages. It is known that in different cells silica NP internalization might involve distinct mechanisms such as, for example (to cite only the most recent contributions), caveolin-initiated macropinocytosis in alveolar epithelial cells (Detampel et al. Sci. Rep. 2022) and LDLR-dependent internalization that does not involve a clathrin-mediated mechanism and depends on the protein-corona adsorbed on silica NP (Francia et al., Nano Lett. 2022). More interestingly, the recent contribution by Wang Q. et al. (Front. Immunol. 2021) indicates that surfactant-associated proteins A and D (SP-A/D) opsonize nanomaterials enhancing their phagocytosis by alveolar macrophages through a mechanism that requires the TLR4 coreceptor CD14. Thus, it is possible that also NM203, the ASNP used in our study, interact with CD14 causing the internalization of the TLR4-CD14 complex. These considerations, along with the references, have been now introduced in the discussion (lines 576-587). We thank the reviewer for arising this topic that was not discussed in the previous version of the manuscript and will be further investigated in the next future.

  Minor:

  1. Line 553 and 576 : no graphical abstract can be seen.

R: The graphical abstract was submitted in the original manuscript and has been now resubmitted. We apologize for this omission, although we don’t know the cause of this problem.

  1. As in THP-1 cells [19], also in MDM ASNP prevented the LPS-dependent increase of GS expression, attributable to the induction of the GLUL gene. Grammar should be checked

R: The sentence has been corrected (lines 591-592). We thank the reviewer for the suggestion.  

  1. Line 567-569: What do you mean “missed skewing ?”, you said GS inhibition stimulates the secretion of pro-inflammatory cytokines and a M1-shift, so I think in this sentence you believed the MDM exposed to ASNP will not be differentiated into M1 type macrophages, but M2 type, right?

R: The sentence has been now re-phrased (see revised discussion, lines 603-608). What we would like to stress is that, although ASNP pre-exposure clearly inhibits the LPS-dependent induction of GS, the synthesis and secretion of the cytokines we tested does not require the expression of the enzyme. This means that the ASNP-dependent inhibition of cytokine secretion observed under LPS challenge is not attributable to the lack of GS protein induction. This observation suggests that these two biological effects of ASNP are mediated by independent mechanisms. How this would impact the MDM phenotypic skewing M1-M2 is a matter of hypothesis. If GS induction represents a determinant of M2 skewing (see refs. 37-38), then ASNP should prevent this polarization. Long term experiments and marker characterization will clarify this interesting issue.

  1. Line 556: RAW 264.7; also in all place in the article.

R: We thank the reviewer for the remark. We have checked and corrected accordingly throughout the manuscript.

  1. Line 106-108: while for all the other experiments MDM were pre-exposed to 10 μg/cm2 of ASNP and then incubated in the presence or in the absence of LPS (1 ng/ml) for the last 24 h. What is the exposure duration for the ASNP? 24 h?

R: In all the experiments macrophages were exposed to ASNP for 48h. For the ASNP+LPS condition, the endotoxin was added after 24h of pre-exposure to ASNP and maintained for further 24h, so that the total period of exposure to ASNP was 48h also in this case. We have described the method in the text (line 107).

Round 2

Reviewer 1 Report

The authors properly addressed to my questions, and add comments in manuscript.

Reviewer 3 Report

The authors have improved the quality of the images and addressed all my concerns in the revision.